# TD3B: Transition-Directed Discrete Diffusion for Allosteric Binder Generation

**Hanqun Cao,**[1,*] **Aastha Pal,**[2,*] **Sophia Tang,**[3] **Yinuo Zhang,**[3,4] **Jingjie Zhang,**[1]
**Pheng-Ann Heng,**[1] **Pranam Chatterjee**[2,3,†]

[1]Department of Computer Science and Engineering, The Chinese University of Hong Kong
[2]Department of Bioengineering, University of Pennsylvania
[3]Department of Computer and Information Science, University of Pennsylvania
[4]Centre for Computational Biology, Duke-NUS Medical School

[*]These authors contributed equally
[†]Corresponding author: pranam@seas.upenn.edu

## Abstract

Protein function is often regulated not by stabilizing a single conformation, but by biasing the direction of transitions between functional states, as in agonist and antagonist modulation. This distinction is especially critical for clinically relevant signalling modules such as G protein–coupled receptors (GPCRs), where therapeutic efficacy depends on how ligands reshape signaling dynamics rather than on equilibrium binding alone. However, most structure-based binder design methods rely on static conformations and lack a representational mechanism for non-reversible, directional effects, making it difficult to systematically distinguish or design agonist versus antagonist behavior. To address this gap, we introduce **T**ransition-**D**irected **D**iscrete **D**iffusion for allosteric **B**inder design (**TD3B**), a sequence-based generative framework that learns latent cellular signaling representations reflecting ligand-induced state transitions. TD3B treats ligand binding as a directional perturbation of a protein's signaling dynamics and integrates a target-aware direction oracle, a soft binding-affinity gate, and amortized fine-tuning of a pre-trained discrete diffusion model. By explicitly optimizing a directional transition control objective, TD3B generates latent representations in which agonist and antagonist effects correspond to distinct cellular signaling trajectories, independent of binding strength. Anonymous code is available at link.

## 1 Introduction

Protein allostery governs regulation and control across diverse biological processes, including signaling, transport, and transcription. Critically, allosteric function is inherently dynamic Motlagh et al. (2014); Weikl & Paul (2014): biological outcomes often arise from biased transitions between functional states rather than stabilization of a single structure Henzler-Wildman & Kern (2007); Cao et al. (2021). This is most apparent in agonist and antagonist modulation, where ligands induce opposite functional effects on the same protein by reshaping transition pathways between macrostates such as activation and inactivation. In these systems, function is defined by directionality—which transitions are promoted or suppressed—rather than on equilibrium occupancy alone. Despite this, most contemporary binder design algorithms, such as RFdiffusion Watson et al. (2023), BindCraft Pacesa et al. (2025), BoltzGen Stark et al. (2025), and GPCR-specialized extensions like RareFoldG-PCR Li et al. (2025), treat proteins as fixed objects and frame design around stabilizing a target structure or interface, implicitly assuming that functional effects are encoded in static structures. While these methods can generate binders or even active agonists, they remain intrinsically tied to equilibrium structural priors and lack a representational mechanism to represent or control transition asymmetry. As a result, they cannot distinguish or systematically modulate agonist versus antagonist behavior, since static structures alone do not encode non-reversible, directional effects. Consequently, binders whose function arises from reshaping kinetic pathways rather than stabilizing endpoints lie outside the representational scope of structure-centric design approaches.

Recent advances in discrete generative modeling have produced high-capacity peptide language models that learn rich representations of peptide sequence space independent of downstream objectives Tang et al. (2025a;c); Chen et al. (2025b;a); Vincoff et al. (2025); Tang et al. (2025b). In particular, masked discrete diffusion language models (MDLMs) such as PepMDLM learn strong unconditional priors over valid peptide sequences, decoupling sequence syntax and diversity from task-specific optimization Tang et al. (2025a;c). Building on this foundation, guidance strategies such as Pep-Tune and TR2-D2 introduce lightweight fine-tuning and objective-guided sampling mechanisms that bias generation toward desired properties without retraining the base model from scratch Tang et al. (2025a;c). These developments suggest a natural separation between representation learning and functional control: a pre-trained diffusion model provides a general representation of peptide sequence space, while task-specific objectives shape how that space is explored. In this work, we leverage this separation by treating directional allosteric control as a guidance objective layered on top of an existing discrete diffusion backbone, rather than as a new generative architecture. **This leads to the central question we address: how should a guidance objective be formulated when the desired biological effect is directional modulation of protein state transitions rather than equilibrium binding affinity?**

In this work, we introduce **T**ransition-**D**irected **D**iscrete **D**iffusion for allosteric **B**inder design (**TD3B**), a discrete generative framework that treats directional allostery as a key design objective. We model binder action through *sequence-conditioned transition operators* over protein macrostates, explicitly accommodating non-reversible, antisymmetric state changes. By incorporating directional supervision, TD3B biases generation toward binders that promote or suppress specific state transitions, reframing binder design as a non-equilibrium control problem rather than a static structure optimization task. This formulation enables the generation of binders that modulate protein function by reshaping transition directionality, aligning learned representations with biologically meaningful signaling behavior.

Our contributions are threefold:

1. **A transition-operator formulation of directional allosteric control.** We formalize binder-mediated allostery as a sequence-conditioned transition operator over protein macrostates, making directionality and non-reversibility explicit modeling targets beyond static or equilibrium-based representations.

2. **A directionally guided generative framework for binder design.** We introduce a discrete generative modeling approach that fine-tunes an existing peptide generator using direction-only supervision, enabling targeted biasing of transition directionality.

3. **Directional binder design beyond static models.** We demonstrate that contrastive, direction-based fine-tuning produces binders that selectively bias agonistic transitions while minimally affecting reverse transitions, capturing functional behaviors.

## 2 REPRESENTING DIRECTIONAL ALLOSTERY AS A TRANSITION LEARNING PROBLEM

We formalize directional allosteric binder design as an amortized objective-guided sequence generation problem, where a pre-trained MDLM is fine-tuned to generate sequences that bias latent representations of protein state transitions in a specified direction. The preliminaries are shown in Appendix B.

### 2.1 DATA AND DIRECTIONAL SUPERVISION

We assume access to a dataset:

$$\mathcal{D} = \{(x^{(n)}, y^{(n)}, a^{(n)})\}_{n=1}^{N}, \tag{1}$$

where $x^{(n)}$ is the target protein sequence, $y^{(n)} \in \mathcal{A}^L$ is a binder sequence, and

$$a^{(n)} \in \{\text{full agonist, partial agonist, antagonist, negative}\} \tag{2}$$

is a categorical functional action label. Negative labels indicate lack of binding and are excluded from directional supervision.

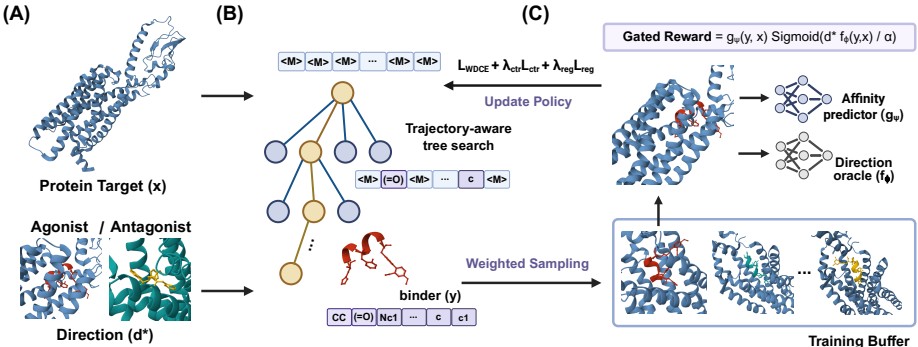

Figure 1: **Overview of the TD3B framework. (A)** TD3B models ligand binding as a directional perturbation of a protein signaling module, enabling virtual control over state transitions in a learned cellular representation. **(B)** Sampling phase: TD3B performs trajectory-aware tree search conditioned on the target and desired direction, generating binder candidates through weighted sampling. **(C)** Finetuning phase: The policy model is updated using samples from the training buffer, guided by the gated reward.

For each target protein, we adopt a two-macrostate abstraction:

$$\mathcal{S} = \{s_{\text{inactive}}, s_{\text{active}}\}. \tag{3}$$

A binder sequence $y$ induces a sequence-conditioned generator $Q^{(y)}$ on $\mathcal{S}$, but neither $Q^{(y)}$ nor its transition rates are observed. Instead, functional labels specify the sign of the induced transition asymmetry:

$$\Delta(y) := Q^{(y)}(s_{\text{inactive}}, s_{\text{active}}) - Q^{(y)}(s_{\text{active}}, s_{\text{inactive}}). \tag{4}$$

We encode supervision using a direction label $d(y) \in \{+1, -1\}$ and a confidence weight $\kappa(y) \in [0, 1]$:

$$d(y) = \begin{cases} +1, & a(y) \in \{\text{full agonist, partial agonist}\}, \\ -1, & a(y) = \text{antagonist}, \end{cases}$$

$$\kappa(y) = \begin{cases} 1, & a(y) = \text{full agonist}, \\ \kappa_{\text{part}}, & a(y) = \text{partial agonist}, \\ 1, & a(y) = \text{antagonist}, \\ 0, & a(y) = \text{negative}, \end{cases} \tag{5}$$

where $\kappa_{\text{part}} \in (0, 1)$ is a hyperparameter reflecting lower confidence in partial agonism.

## 2.2 DIRECTION ORACLE

We introduce a Direction Oracle $f_\phi : \mathcal{A}^L \times \mathcal{X} \to [-1, 1]$, parameterized by $\phi$, that predicts the direction of transition bias. The predicted direction is defined as:

$$\hat{d}(y) = \text{sign}(f_\phi(y, x)). \tag{6}$$

Given a target protein sequence $x$ and a peptide binder $y$, the representations are obtained through pre-trained encoders:

$$\mathbf{h}_x = \text{Pool}(\mathcal{E}_x(x)), \quad \mathbf{h}_y = \text{Pool}(\mathcal{E}_y(y)), \tag{7}$$

where $\mathbf{h}_x \in \mathbb{R}^d$ and $\mathbf{h}_y \in \mathbb{R}^d$ denote the pooled embeddings for the target and binder. The fused representation is computed via a gated mechanism followed by an MLP:

$$\mathbf{z} = \mathbf{g} \odot \mathbf{h}_x + (1 - \mathbf{g}) \odot \mathbf{h}_y, \quad f_\phi(y, x) = \text{MLP}(\mathbf{z}), \tag{8}$$

where $\mathbf{g} = \sigma(\mathbf{W}_g[\mathbf{h}_x; \mathbf{h}_y] + \mathbf{b}_g)$ is a learned gating vector and $\odot$ denotes element-wise multiplication.

The oracle minimizes a weighted binary classification loss:

$$\mathcal{L}_{\text{dir}}(\phi) = \mathbb{E}_{(x,y,d)\sim\mathcal{D}} \left[ \kappa(y) \log(1 + \exp(-d \cdot f_\phi(y, x))) \right],$$

where $d \in \{-1, +1\}$ denotes the ground-truth direction and $\kappa(y)$ is a sample-dependent weight.

## 2.3 Contrastive Directional Representation

Let $h_\theta(y) \in \mathbb{R}^m$ denote a sequence representation extracted from the MDLM by mean-pooling the final-layer hidden states across all sequence positions. To enforce separation between directional classes in representation space, we define positive and negative index sets:

$$\mathcal{P} = \{(i,j) : d(y_i) = d(y_j), \ \kappa(y_i)\kappa(y_j) > 0\}, \tag{9}$$
$$\mathcal{N} = \{(i,j) : d(y_i) \neq d(y_j), \ \kappa(y_i)\kappa(y_j) > 0\}. \tag{10}$$

A margin-based contrastive loss is defined as:

$$
\begin{aligned}
\mathcal{L}_{\mathrm{ctr}}(\theta) = &\sum_{(i,j) \in \mathcal{P}} \|h_\theta(y_i) - h_\theta(y_j)\|_2^2 \\
&+ \sum_{(i,j) \in \mathcal{N}} \max\left(0, \ m - \|h_\theta(y_i) - h_\theta(y_j)\|_2\right)^2,
\end{aligned}
\tag{11}
$$

where $m > 0$ is a margin hyperparameter. Negative (non-binding) samples are excluded from this loss.

## 2.4 Incorporating Target Binding Affinity via Gating

Directional allosteric control is only meaningful for sequences that actually bind to the target protein. To ensure that directional supervision is applied to plausible binders, we incorporate a pre-trained peptide-protein affinity predictor as a soft gate within the reward function.

Let

$$g_\psi(y, x) \in [0, 1] \tag{12}$$

denote a pre-trained affinity model that predicts the probability that peptide $y$ binds target protein $x$.

Given a desired direction $d^\star \in \{+1, -1\}$, we define the gated reward as:

$$R(y; d^\star, x) = g_\psi(y, x) \cdot \sigma\left(\frac{d^\star \cdot f_\phi(y, x)}{\tau}\right), \tag{13}$$

where $\sigma$ denotes the sigmoid function and $\tau$ is a temperature coefficient. This formulation ensures that sequences predicted not to bind contribute negligible reward regardless of their directional score, while sequences predicted to bind are ranked according to their directional effect. Crucially, binding affinity acts as a gate to filter implausible candidates, not as a quantity to be maximized—stronger binders do not necessarily induce desired state changes.

The resulting reward-tilted target distribution becomes:

$$p^\star(y \mid d^\star, x) \ \propto \ p_{\theta_0}(y) \exp\left(\frac{R(y; d^\star, x)}{\alpha}\right), \tag{14}$$

where $\alpha > 0$ controls the strength of deviation from the pre-trained prior.

Following the trajectory-level importance weighting framework from Section B.3, the unnormalized log importance weight for a trajectory $\boldsymbol{X}_{0:T}$ with final sequence $y = \boldsymbol{X}_T$ is:

$$
\begin{aligned}
\log \tilde{w}(\boldsymbol{X}_{0:T}) = &\frac{R(y; d^\star, x)}{\alpha} \\
&+ \sum_{t=1}^{T} \sum_{\ell : \boldsymbol{X}_{t-1}^\ell \neq \boldsymbol{X}_t^\ell} \log \frac{p_{\theta_0}(\boldsymbol{X}_{t-1}^\ell \mid \boldsymbol{X}_t^{\mathrm{UM}})}{p_{\bar{\theta}}(\boldsymbol{X}_{t-1}^\ell \mid \boldsymbol{X}_t^{\mathrm{UM}})},
\end{aligned}
\tag{15}
$$

where $p_{\theta_0}$ and $p_{\bar{\theta}}$ denote the pre-trained model and proposal policy. At inference, a target protein $x$ and direction $d^\star \in \{+1, -1\}$ (agonist/antagonist) are provided. These inputs condition the reward function alone, keeping the generative backbone target-agnostic.

## 2.5 Amortized Fine-Tuning Objective

Direct sampling from $p^\star$ is intractable. We therefore learn a new parameterization $p_\theta(y)$ via amortized fine-tuning. Let $\mathcal{B}$ denote a replay buffer of sequences approximately sampled from $p^\star$ via tree search. The WDCE objective is:

$$\mathcal{L}_{\mathrm{WDCE}}(\theta) = \mathbb{E}_{y \sim \mathcal{B}} \, \mathbb{E}_{t, y_t \sim q_t(\cdot | y)} \Big[ w(y) \sum_{\ell : (y_t)_\ell = [\mathrm{MASK}]} - \log p_\theta(y_\ell \mid (y_t)_{\mathrm{UM}}, t) \Big], \tag{16}$$

where the importance weight $w(y)$ is computed as:

$$w(y) = \frac{\exp(R(y; d^\star, x)/\alpha)}{\sum_{y' \in \mathcal{B}} \exp(R(y'; d^\star, x)/\alpha)}. \tag{17}$$

This softmax normalization over the buffer converts unnormalized log-weights into valid importance weights. Samples with $\kappa(y) = 0$ (non-binders) contribute no gradient.

To prevent collapse and preserve the pre-trained prior, we include a regularization term:

$$\mathcal{L}_{\mathrm{reg}}(\theta) = \mathrm{KL}(p_\theta \, \| \, p_{\theta_0}). \tag{18}$$

The full fine-tuning objective is:

$$\min_\theta \; \big\{ \mathcal{L}_{\mathrm{WDCE}}(\theta) + \lambda_{\mathrm{ctr}} \, \mathcal{L}_{\mathrm{ctr}}(\theta) + \lambda_{\mathrm{reg}} \, \mathcal{L}_{\mathrm{reg}}(\theta) \big\}, \tag{19}$$

where $\lambda_{\mathrm{ctr}}, \lambda_{\mathrm{reg}} \geq 0$ are hyperparameters. At generation time, users specify a target protein $x$ and a desired direction $d^\star \in \{+1, -1\}$ corresponding to agonist or antagonist behavior.

## 2.6 Design Task

The directional allosteric design task is formally defined as:

> **Design Task:** Given a desired direction $d^\star$, generate binder sequences to probe and control the directionality of signaling within a learned cellular model.

This formulation treats directionality of state transitions as the primary generative objective and defines a fully amortized procedure for incorporating coarse functional supervision into discrete sequence generation.

# 3 Results

We designed virtual perturbation experiments to test whether modeling binder action as a sequence-conditioned transition operator captures forms of allosteric control that are inaccessible to equilibrium- and structure-centric design methods. Our evaluation addresses three questions: (1) whether directionally fine-tuned generators induce non-reversible transition behavior; (2) whether directional control can be achieved independently of binding affinity; and (3) whether the framework supports targeted control over specific transition directions rather than global perturbations.

## 3.1 Experimental Settings

**Dataset.** We curated data from the IUPHAR/BPS Guide to Pharmacology database Harding et al. (2026), classifying entries as agonist or antagonist based on bioactivity. Interacting residues on ligands were extracted using PeptiDerive Sedan et al. (2016) and converted to canonical SMILES representations for downstream processing. To balance agonist and antagonist samples, we generated synthetic antagonist ligands via RFDiffusion Watson et al. (2023) for targets with known agonists. After redundancy removal using MMseqs2 Steinegger & Söding (2017), the final dataset comprises 1,446 training and 336 held-out test samples.

The data split for TD3B training, validation, and test sets follows that of the Direction Oracle. Within the Direction Oracle training set, we further partitioned the data into training and validation subsets at an 8:1 ratio based on clustering. We filtered out binder sequences with residue counts outside the range of [16, 128], as well as targets containing only a single direction type (agonist or antagonist), to prevent direction bias toward specific targets. This preprocessing yields 130, 34, and 88 bidirectional target-binder pairs for the training, validation, and test sets, respectively.

**Evaluation Metrics.**    For the Direction Oracle classification performance, we report Accuracy, Precision, Recall, and F1 Score to evaluate discriminative capability. To assess directional accuracy, inter-direction transitions, and designed binder affinity, we introduce direction-specific metrics for both agonist and antagonist modes: Affinity ($d^* = 1$), Affinity ($d^* = -1$), Direction Accuracy ($d^* = 1$), and Direction Accuracy ($d^* = -1$). Additionally, we report the gated reward to evaluate overall model training effectiveness.

**Setup.**    We adopt the pre-trained MDLM weights from PepTune Tang et al. (2025a) as our base model. During finetuning, we employ tree search for buffer generation following TR2-D2 Tang et al. (2025c), regenerating binder data for multiple targets every $k$ iterations with a first-in-first-out buffer to mitigate catastrophic forgetting. Binding affinity is estimated via a pre-trained predictor Tang et al. (2025a); Zhang et al. (2026), trained on the PepLand dataset Zhang et al. (2025) to produce a continuous, normalized affinity score (combining $K_d$, $K_i$, and $IC_{50}$), where higher values indicate stronger binding and a one-unit increase corresponds to an approximate tenfold change in binding strength. For the Direction Oracle, protein target sequences are encoded using ESM2 Lin et al. (2023), and binder sequences are tokenized using the SPE tokenizer Tang et al. (2025c). During generation, we apply Algorithm 2 for weighted sampling, generating 8 samples per specified direction for each target.

## 3.2   Direction Oracle for Protein-Peptide Binding

To ensure sufficient exploration space for both tree search and model training, we first demonstrate that the Direction Oracle can accurately identify the directionality of binders across diverse targets and varying sequence lengths, thereby providing reliable guidance during training. As shown in Table 1, the Direction Oracle achieves strong classification performance across all metrics.

Table 1: Binary classification performance of the Direction Oracle.

|  | Accuracy | Precision | Recall | F1 |
|---|---|---|---|---|
| Dir. Oracle | 0.93 | 0.90 | 0.91 | 0.90 |

## 3.3   Assessing Directional Asymmetry of Learned Cellular Transition Operators

Binding affinity is a prerequisite for characterizing agonists and antagonists Brunton et al. (2018) To validate TD3B, we compared the predicted binding affinity of TD3B-generated samples against those designed by structure-based RFDiffusion Watson et al. (2023). Figure 2A shows TD3B achieves higher predicted normalized affinity, demonstrating the effectiveness of gated reward fine-tuning.

We evaluated the directional behavior of peptides generated under agonist- versus antagonist-directed finetuning by measuring samples from the pre-trained (unconditioned) generator and TD3B using the Direction Oracle. As shown in Figure 2B, the pre-trained model generates predominantly agonist-biased binders with low confidence and lacks directional control. In contrast, TD3B produces distributions with higher confidence across both directions and enables explicit control over transition directionality, though we note asymmetry remains (see Table 2).

## 3.4   Comparison to Static and Predictive Baselines

We investigated whether TD3B can overcome the directional symmetry between activation and inhibition states. Since pre-trained discrete diffusion models do not support direction as a conditional

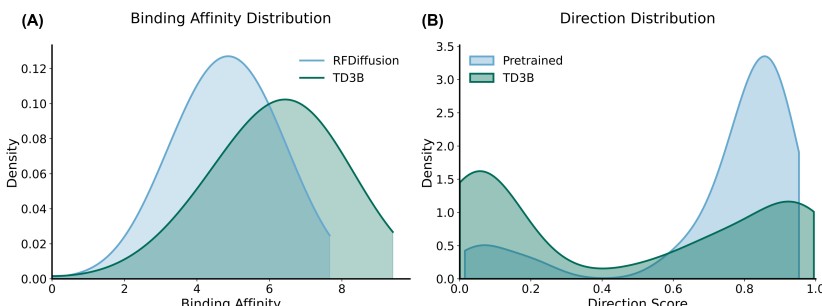

Figure 2: Direction and affinity distribution of TD3B. **(A)** Binding affinity comparison vs. RFDiffusion. **(B)** Direction comparison vs. pre-trained PepTune.

Table 2: Comparison of model performance across various metrics. Best results are in bold. CG: Classifier Guidance; SMC: Sequential Monte Carlo; TDS: Twisted Diffusion Sampler Wu et al. (2023).

| Model | Affinity $(d^* = 1)$ | Affinity $(d^* = -1)$ | Dir. Acc. $(d^* = 1)$ | Dir. Acc. $(d^* = -1)$ | Gated Reward |
|---|---|---|---|---|---|
| Pre-trained | 4.80 | 4.80 | 0.84 | 0.18 | 2.1 |
| CG | 4.80 | 4.80 | 0.82 | 0.20 | 2.2 |
| SMC | 4.83 | 4.83 | 0.89 | 0.64 | 3.2 |
| TDS | 4.86 | 4.74 | 0.90 | 0.16 | 2.9 |
| PepTune | 4.94 | 4.94 | 0.88 | 0.28 | 2.8 |
| TR2-D2 | 5.87 | 5.95 | 0.32 | 0.82 | 3.6 |
| TD3B w/o $\mathcal{L}_{\mathrm{ctr}}$ | 5.88 | 6.05 | 0.57 | 1.00 | 4.1 |
| TD3B w/o $\mathcal{L}_{\mathrm{reg}}$ | 5.82 | 5.46 | **0.95** | 0.30 | 3.8 |
| **TD3B** | **5.92** | **6.30** | 0.66 | **1.00** | **5.2** |

input, we compare training-free guided diffusion approaches against tree-search-based finetuning. As shown in Table 2, TD3B achieves state-of-the-art performance on the gated reward, demonstrating its effectiveness for direction-specific generation. The directional accuracy shows asymmetry between agonist and antagonist modes, likely reflecting the stronger antagonist signal in the training data and the pre-trained model's inherent agonist bias. Finetuning-based methods achieve higher binding affinity while maintaining better balance across directions compared to guidance-only approaches. Compared to TR2-D2, TD3B employs weighted sampling to exploit high-potential samples and contrastive loss to separate sample distributions in latent space, enabling directional understanding beyond iterative optimization alone.

## 3.5 Targeted Control of Transition Asymmetry

We next assessed TD3B's targeted control over specific transitions without global dynamic perturbation. Conditioning on $d^\star \in \{+1, -1\}$, we verified that generated binders selectively bias transitions while maintaining high affinity. Table 3 shows de novo TD3B binders outperform length-matched wild-type references in affinity across all directions. Success rate—the fraction of binders achieving both (i) superior predicted affinity to wild-type and (ii) correct Direction Oracle classification—reaches 61% and 100% for forward and backward transitions, respectively. This confirms directionality as a controllable objective rather than an incidental outcome of binding optimization.

Table 3: Targeted control: Evaluation of functional specificity across transition objectives.

| Design Objective ($d^*$) | Affinity (WT) | Affinity (Transition) | Success Rate |
|---|---|---|---|
| Forward Transition ($d^* = 1$) | 4.66 | 5.81 | 0.61 |
| Reverse Transition ($d^* = -1$) | 4.99 | 6.06 | 1.00 |

## 3.6 Case Studies

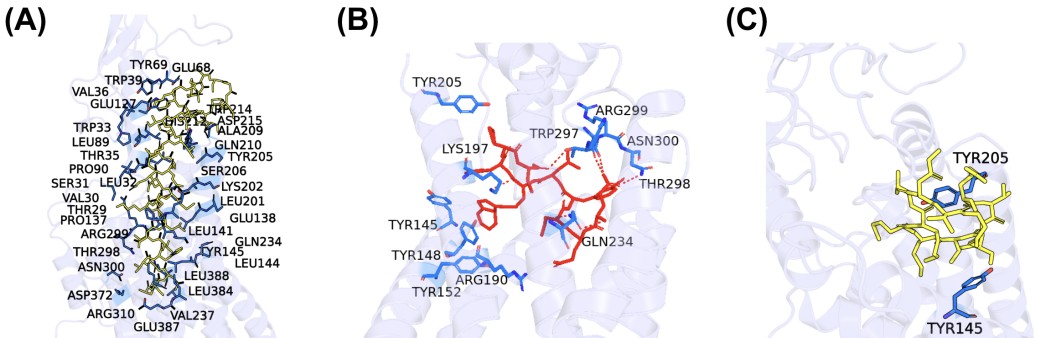

Figure 3: Evaluation of TD3B on GLP-1R. **(A)** Existing GLP-1 peptide hormone bound to GLP-1R. **(B)** TD3B-designed agonist bound GLP-1R. **(C)** TD3B-designed antagonist bound GLP-1R.

We fine-tuned TD3B on two key GPCR targets (GLP1R and TAAR1) and selected top-ranked candidates for each. Complex structures were predicted using AlphaFold3, followed by binding-site detection and scoring with an external classifier. We first focused on GLP-1R, a clinically important metabolic receptor revolutionary for obesity, weight loss, and type 2 diabetes treatment where agonist activity is the primary therapeutic mechanism Moiz et al. (2025). As reference, Figure 3A shows the AF3-predicted structure of GLP-1R with all 37 interacting residues at the binding interface of the endogenous GLP-1 hormone. TD3B-designed agonists engage key activation residues including Tyr148, Tyr152, Arg190, Lys197, Tyr205, Gln234, Trp297, Thr298, Arg299, and Asn300 Liao & Tzen (2022), with Arg299 and Asn300 being essential for full agonist activity Lei et al. (2018) (Figure 3B). In contrast, TD3B-designed antagonists (Figure 3C) lack interactions with Arg299 and Asn300, consistent with their inability to activate the receptor. These results demonstrate that TD3B can design agonist and antagonist binders that selectively engage or avoid critical activation residues on the same receptor. More cases are shown in Appendix E.

## 3.7 Ablation Studies

Finally, we validate the effectiveness of each loss component. As shown in Table 2, removing either $\mathcal{L}_{ctr}$ or $\mathcal{L}_{reg}$ results in performance degradation. Without $\mathcal{L}_{ctr}$, direction accuracy drops substantially, as the overlapping latent space potentially undermines the model's ability to discriminate between activation and inhibition pathways. Moreover, without $\mathcal{L}_{reg}$, agonist accuracy increases to 0.95, yet generates biologically implausible sequences with impaired binding affinity, while antagonist mode nearly collapses (Acc 0.30), indicating that the antagonist feature space is more unstable to degradation during unconstrained finetuning.

## 4 Discussion

We introduce a generative framework, **TD3B**, that treats allosteric binder design as control over sequence-conditioned transition operators rather than optimization of static states or equilibrium energies. By making directional asymmetry and non-reversibility explicit modeling targets, the formulation captures functional effects that are not well-modeled by structure-centric design algorithms or predictive dynamics models. While we instantiate the framework with sequence-based binders, the transition-operator perspective is modality-agnostic and provides a general foundation for generative design in settings where function arises from non-equilibrium state shifts.

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

# Appendix

## A    ALGORITHMS

---

**Algorithm 1 Direction-Only Amortized Fine-Tuning of an MDLM**

---

1: **Input:** pre-trained MDLM $p_{\theta_0}(y)$
2:          directional dataset $\mathcal{D} = \{(y^{(n)}, a^{(n)})\}_{n=1}^N$
3:          direction oracle $f_\phi(y)$
4:          replay buffer $\mathcal{B} \leftarrow \emptyset$
5: **Hyperparameters:** learning rate $\eta$, temperature $\alpha$, contrastive weight $\lambda_{\mathrm{ctr}}$, KL weight $\lambda_{\mathrm{reg}}$
6: **while** not converged **do**
7:     Sample minibatch $\{(y, a)\}$ from $\mathcal{D}$
8:     Compute direction labels $d(y) \in \{+1, -1\}$ and weights $\kappa(y)$
9:     Update direction oracle parameters $\phi$ using $\mathcal{L}_{\mathrm{dir}}$
10:                        ▷ *Populate replay buffer with direction-aligned samples*
11:     Sample candidate sequences $\{\tilde{y}_k\}_{k=1}^M$ from $p_\theta(y)$
12:     **for** each $\tilde{y}_k$ **do**
13:        Compute direction score $S(\tilde{y}_k) = \sigma(d^\star \cdot f_\phi(\tilde{y}_k)/\tau)$
14:        Set importance weight $w(\tilde{y}_k) \propto \exp(S(\tilde{y}_k)/\alpha)$
15:     **end for**
16:     Add weighted samples $\{(\tilde{y}_k, w(\tilde{y}_k))\}$ to replay buffer $\mathcal{B}$
17:                      ▷ *Fine-tune MDLM using WDCE and contrastive objectives*
18:     Compute $\mathcal{L}_{\mathrm{WDCE}}(\theta)$ using samples from $\mathcal{B}$
19:     Compute $\mathcal{L}_{\mathrm{ctr}}(\theta)$ on non-negative samples
20:     Compute regularization $\mathcal{L}_{\mathrm{reg}}(\theta) = \mathrm{KL}(p_\theta \| p_{\theta_0})$
21:     Update $\theta \leftarrow \theta - \eta \nabla_\theta (\mathcal{L}_{\mathrm{WDCE}} + \lambda_{\mathrm{ctr}}\mathcal{L}_{\mathrm{ctr}} + \lambda_{\mathrm{reg}}\mathcal{L}_{\mathrm{reg}})$
22: **end while**
23: **return** fine-tuned generator $p_{\theta^*}(y)$ and oracle $f_{\phi^*}$

---

---

**Algorithm 2 Sampling Directional Allosteric Binders**

---

1: **Input:** fine-tuned MDLM $p_{\theta^*}(y)$
2:          desired direction $d^\star \in \{+1, -1\}$
3:          direction oracle $f_{\phi^*}(y)$
4:          number of candidates $M$
5: Sample candidate binders $\{y_m\}_{m=1}^M$ i.i.d. from $p_{\theta^*}(y)$
6: **for** $m = 1$ **to** $M$ **do**
7:     Compute direction score $S(y_m) = d^\star \cdot f_{\phi^*}(y_m)$
8:     Set weight $w_m \propto \exp(S(y_m))$
9: **end for**
10: Resample $y \sim \mathrm{Categorical}(\{w_m\}_{m=1}^M)$
11: **return** $y$                       ▷ *directionally biased binder*

---

# B PRELIMINARIES

This section reviews the generative modeling primitives and dynamical abstractions underlying our approach. We describe masked discrete diffusion language models (MDLMs) as a general-purpose backbone for sequence generation Sahoo et al. (2024); Austin et al. (2021); Shi et al. (2024), introduce amortized objective-guided fine-tuning as a mechanism for incorporating external objectives Tang et al. (2025c), and define a coarse-grained dynamical representation of protein function in terms of macrostates and transition operators. These components are standard or well-established; we defer the specifics of our proposed method to Section 2.

## B.1 DISCRETE SEQUENCE SPACES

Let $\mathcal{A}$ denote a finite alphabet and $\mathcal{A}^L$ the space of length-$L$ sequences. In this work, sequences correspond to binders, and the formulation applies to arbitrary sequence-based binders, with peptides serving as a concrete instantiation. We write $y \in \mathcal{A}^L$ for a clean sequence and use $x$ to denote target protein sequences.

## B.2 MASKED DISCRETE DIFFUSION LANGUAGE MODELS

Masked discrete diffusion language models (MDLMs) define generative processes over discrete sequences by progressively denoising corrupted inputs. A forward noising process $q_t(x_t \mid x)$ maps a clean sequence $x$ to a partially corrupted version $x_t$ at time $t \in [0, 1]$, typically by masking tokens according to a time-dependent schedule. The reverse process is parameterized by a neural network that predicts the distribution of masked tokens conditioned on the unmasked context and time.

Training proceeds by minimizing a denoising cross-entropy objective:

$$
\mathcal{L}_{\mathrm{DCE}}(\theta, \boldsymbol{x}) = \mathbb{E}_{t \sim \mathcal{U}(0,1)} \left[ \frac{1}{t} \mathbb{E}_{q_t(\tilde{\boldsymbol{x}}_t | \boldsymbol{x})} \right.
$$

$$
\left. \sum_{\ell : \boldsymbol{x}_t^\ell = \boldsymbol{M}} -\log p_\theta \left( \boldsymbol{x}^\ell \mid \boldsymbol{x}_t^{\mathrm{UM}}, t \right)_{\boldsymbol{x}^\ell} \right], \tag{20}
$$

where $(x_t)_{\mathrm{UM}}$ denotes the unmasked tokens. Once trained, an MDLM defines an unconditional distribution over valid sequences and can be sampled by iteratively denoising from a fully masked input. Importantly, the MDLM captures the combinatorial structure of sequence space independently of any downstream task.

## B.3 AMORTIZED OBJECTIVE-GUIDED FINE-TUNING OF MDLMS

Let $p_{\theta_0}(x)$ denote a pre-trained MDLM defining an unconditional distribution over sequences. Objective-guided sequence design seeks to bias this distribution toward sequences that score highly under an external objective $S(x)$, while preserving the structural prior learned by $p_{\theta_0}$.

Amortized fine-tuning methods, such as TR2-D2 Tang et al. (2025c), achieve this by learning a new parameterization $p_\theta(x)$ that approximates a reward-tilted target distribution:

$$
p^*(x) \propto p_{\theta_0}(x) \exp\left( \frac{S(x)}{\alpha} \right), \tag{21}
$$

where $\alpha > 0$ controls the strength of deviation from the pre-trained prior. This formulation defines an implicit energy-based reweighting of the base model without modifying the corruption process or denoising schedule.

Since direct sampling from $p^*$ is intractable, amortized fine-tuning optimizes $\theta$ via a weighted denoising cross-entropy (WDCE) objective. Let $\boldsymbol{X}_{0:T}$ denote a denoising trajectory with final

sequence $\boldsymbol{x} = \boldsymbol{X}_T$. The fine-tuning objective is:

$$\mathcal{L}_{\text{WDCE}}(\theta) = \mathbb{E}_{p_{\text{target}}(\boldsymbol{x})}[\mathcal{L}_{\text{DCE}}(\theta; \boldsymbol{x})] \tag{22}$$

$$= \mathbb{E}_{\boldsymbol{X}_{0:T} \sim \mathbb{P}^\star}[\mathcal{L}_{\text{DCE}}(\theta; \boldsymbol{X}_T)] \tag{23}$$

$$= \mathbb{E}_{\boldsymbol{X}_{0:T} \sim \mathbb{P}^v}\left[ \underbrace{\frac{d\mathbb{P}^\star}{d\mathbb{P}^v}(\boldsymbol{X}_{0:T})}_{w(\boldsymbol{X}_{0:T})} \mathcal{L}_{\text{DCE}}(\theta; \boldsymbol{X}_T) \right], \tag{24}$$

where $\mathbb{P}^v$ is a proposal distribution over trajectories and $w(\boldsymbol{X}_{0:T})$ is the importance weight. Under the assumption that the trajectory distribution factorizes and the reward depends only on the final sequence, the trajectory-level importance weight decomposes as:

$$w(\boldsymbol{X}_{0:T}) \propto \exp\left( \frac{S(\boldsymbol{X}_T)}{\alpha} \right) \cdot \prod_t \frac{p_{\theta_0}(\boldsymbol{X}_{t-1} \mid \boldsymbol{X}_t)}{p_{\bar{\theta}}(\boldsymbol{X}_{t-1} \mid \boldsymbol{X}_t)}, \tag{25}$$

where the first term captures the reward contribution and the second term accounts for the mismatch between the pre-trained and proposal transition kernels.

This procedure internalizes objective $S$ into the model's sampling distribution, minimizing the need for extensive inference-time search. While lightweight methods like best-of-$M$ selection can still refine quality, this amortized initialization is notably more efficient than guidance-only approaches. Fine-tuning updates only the denoising conditionals, leaving the diffusion and corruption schedules intact.

## B.4 PROTEIN MACROSTATES AND COARSE-GRAINED STATE SHIFTS

To reason about allosteric function, we adopt a coarse-grained description of protein state shifts. Let

$$\mathcal{S} = \{s_1, \ldots, s_K\} \tag{26}$$

denote a finite set of macrostates corresponding to functionally distinct configurations, such as inactive/active or closed/open. In the absence of a binder, protein state shifts are modeled as a continuous-time Markov chain (CTMC) with generator:

$$Q_0 : \mathcal{S} \times \mathcal{S} \to \mathbb{R}, \qquad Q_0(s_i, s_i) = -\sum_{j \neq i} Q_0(s_i, s_j). \tag{27}$$

This abstraction captures state-to-state transition behavior without committing to atomistic trajectories or detailed kinetic models.

## B.5 SEQUENCE-CONDITIONED TRANSITION OPERATORS

A binder sequence $y \in \mathcal{A}^L$ may alter the transition structure of proteins. We represent this effect by a sequence-conditioned generator:

$$Q^{(y)} = Q_0 + \Delta Q^{(y)}, \tag{28}$$

where $\Delta Q^{(y)}$ denotes a sequence-dependent perturbation of transition rates. No symmetry or reversibility is assumed. In general,

$$Q^{(y)}(s_i, s_j) \neq Q^{(y)}(s_j, s_i), \tag{29}$$

and the resulting dynamics need not satisfy detailed balance:

$$\pi^{(y)}(s_i)Q^{(y)}(s_i, s_j) \neq \pi^{(y)}(s_j)Q^{(y)}(s_j, s_i), \tag{30}$$

While the stationary distribution $\pi^{(y)}$ exists for finite irreducible CTMCs, it does not imply reversibility; thus, these generators lack a scalar energy gradient representation.

For any ordered state pair $(s_i, s_j)$, we define the directional asymmetry induced by a sequence $y$ as:

$$\Delta_{ij}(y) := Q^{(y)}(s_i, s_j) - Q^{(y)}(s_j, s_i). \tag{31}$$

This quantity captures the net bias of transitions between macrostates. Directional allosteric effects correspond to consistent signs of $\Delta_{ij}(y)$ for selected transitions. We emphasize that neither the absolute magnitude of $\Delta_{ij}(y)$ nor the full generator $Q^{(y)}$ is assumed to be observable; only coarse directional information may be available in practice.

The concepts introduced define the modeling primitives throughout this paper. We next specify how these components combine into a concrete generative learning problem.

## C  THEORETICAL PROOFS

This appendix records basic guarantees for TD3B. We (i) justify exponential tilting as the unique solution of a KL-regularized improvement objective, (ii) characterize the population-optimal Direction Oracle under weighted logistic risk, (iii) relate weighted denoising cross-entropy to fitting a target (tilted) distribution, (iv) bound the effect of oracle approximation error on the induced tilted distribution, and (v) state a simple separability consequence of zero contrastive loss.

### C.1  NOTATION AND SETUP

We adopt notation from the main text.

- $\mathcal{Y} := \mathcal{A}^L$ denotes the finite sequence space.
- $p_0(y)$ denotes a fixed base distribution on $\mathcal{Y}$ (e.g., the pre-trained MDLM distribution $p_{\theta_0}$).
- $S : \mathcal{Y} \to \mathbb{R}$ denotes a score (objective) and $\alpha > 0$ a temperature.
- The (reward-)tilted distribution is

$$p^\star(y) := \frac{p_0(y) \exp(S(y)/\alpha)}{Z}, \qquad Z := \sum_{y' \in \mathcal{Y}} p_0(y') \exp(S(y')/\alpha). \qquad (32)$$

- For the direction task, we use labels $d(y) \in \{+1, -1\}$, confidence weights $\kappa(y) \in [0, 1]$, and an oracle $f_\phi : \mathcal{Y} \to \mathbb{R}$. The direction-only score used for design is $S(y; d^\star) = d^\star f_\phi(y)$ for $d^\star \in \{+1, -1\}$.
- For distributions $P, Q$ on $\mathcal{Y}$ we write $\|P - Q\|_{\mathrm{TV}}$ for total variation and $\mathrm{KL}(P\|Q)$ for Kullback–Leibler divergence.

### C.2  EXPONENTIAL TILTING AS KL-REGULARIZED IMPROVEMENT

**Theorem C.1** (Exponential tilting solves KL-regularized improvement). *Fix a base distribution $p_0$ on $\mathcal{Y}$ and a score $S : \mathcal{Y} \to \mathbb{R}$. Consider the optimization problem over distributions $q$ on $\mathcal{Y}$:*

$$\max_{q \in \Delta(\mathcal{Y})} \left\{ \mathbb{E}_{Y \sim q}[S(Y)] - \alpha \, \mathrm{KL}(q\|p_0) \right\}, \qquad \alpha > 0. \qquad (33)$$

*Then the unique maximizer is the tilted distribution $p^\star$ in equation 32.*

*Proof.* Since $\mathcal{Y}$ is finite, equation 33 is a strictly concave optimization problem in $q$. Introduce a Lagrange multiplier $\lambda$ for the constraint $\sum_y q(y) = 1$. The Lagrangian is

$$\mathcal{J}(q, \lambda) = \sum_y q(y) S(y) - \alpha \sum_y q(y) \log \frac{q(y)}{p_0(y)} + \lambda \left( \sum_y q(y) - 1 \right). \qquad (34)$$

Taking derivatives with respect to $q(y)$ and setting to zero gives

$$S(y) - \alpha \left( \log q(y) - \log p_0(y) + 1 \right) + \lambda = 0, \qquad (35)$$

so $\log q(y) = \log p_0(y) + S(y)/\alpha + c$ for a constant $c$. Hence

$$q(y) \propto p_0(y) \exp(S(y)/\alpha), \qquad (36)$$

and normalization yields equation 32. Strict concavity implies uniqueness. $\square$

**Corollary C.1** (Limiting cases). *Let $p^\star$ be defined by equation 32. Then:*

1. *As $\alpha \to \infty$, $p^\star \to p_0$ in total variation.*

2. *As $\alpha \to 0^+$, $p^\star$ concentrates on $\arg\max_{y \in \mathcal{Y}} S(y)$ (within the support of $p_0$).*

*Proof.* Write $p^\star(y) = p_0(y) \exp(S(y)/\alpha)/Z$. As $\alpha \to \infty$, $\exp(S(y)/\alpha) \to 1$ uniformly, so $Z \to 1$ and $p^\star \to p_0$. As $\alpha \to 0^+$, the normalization is dominated by maximizers of $S$ because $\exp(S(y)/\alpha)$ is a softmax with temperature $\alpha$. $\square$

**Proposition C.1** (Relative odds under tilting). *For any $y_1, y_2 \in \mathcal{Y}$ with $p_0(y_1), p_0(y_2) > 0$,*

$$\frac{p^\star(y_1)}{p^\star(y_2)} = \frac{p_0(y_1)}{p_0(y_2)} \exp\left(\frac{S(y_1) - S(y_2)}{\alpha}\right). \tag{37}$$

*In particular, if $p_0(y_1) = p_0(y_2)$ then the odds ratio depends only on $S(y_1) - S(y_2)$.*

*Proof.* Immediate from the definition equation 32 since the normalizer $Z$ cancels. $\square$

**Binary direction specialization.** For $S(y; d^\star) = d^\star f(y)$ and $d^\star \in \{+1, -1\}$,

$$\frac{p^\star(y \mid +1)}{p^\star(y \mid -1)} = \frac{Z_-}{Z_+} \exp\left(\frac{2f(y)}{\alpha}\right), \qquad Z_\pm := \sum_y p_0(y) \exp(\pm f(y)/\alpha). \tag{38}$$

Thus larger oracle score $f(y)$ implies larger posterior odds of the $+1$-tilt relative to the $-1$-tilt.

## C.3 POPULATION OPTIMALITY OF THE DIRECTION ORACLE

We formalize the direction oracle as a weighted logistic risk minimizer.

**Theorem C.2** (Bayes-optimal oracle under weighted logistic loss). *Let $(Y, D)$ be a random pair where $Y \in \mathcal{Y}$ and $D \in \{+1, -1\}$. Let $\kappa : \mathcal{Y} \times \{+1, -1\} \to [0, \infty)$ be a measurable weight (in TD3B, $\kappa$ encodes down-weighting of partial agonists and exclusion of non-binders). Consider minimizing*

$$\mathcal{R}(f) := \mathbb{E}[\kappa(Y, D) \log(1 + \exp(-Df(Y)))] \tag{39}$$

*over all functions $f : \mathcal{Y} \to \mathbb{R}$. Define*

$$\eta_+(y) := \mathbb{E}[\kappa(Y, D)\mathbf{1}\{D = +1\} \mid Y = y], \qquad \eta_-(y) := \mathbb{E}[\kappa(Y, D)\mathbf{1}\{D = -1\} \mid Y = y]. \tag{40}$$

*If $\eta_+(y) + \eta_-(y) > 0$, the pointwise minimizer satisfies*

$$f^\star(y) = \log \frac{\eta_+(y)}{\eta_-(y)}. \tag{41}$$

*Proof.* Fix $y \in \mathcal{Y}$ and write the conditional risk (up to an additive constant independent of $f(y)$) as

$$r_y(u) = \eta_+(y) \log(1 + \exp(-u)) + \eta_-(y) \log(1 + \exp(u)), \qquad u := f(y). \tag{42}$$

This is strictly convex in $u$. Differentiate and set to zero:

$$r_y'(u) = -\eta_+(y) \frac{1}{1 + \exp(u)} + \eta_-(y) \frac{\exp(u)}{1 + \exp(u)} = 0. \tag{43}$$

Multiplying by $1 + \exp(u)$ gives $-\eta_+(y) + \eta_-(y) \exp(u) = 0$, hence $\exp(u) = \eta_+(y)/\eta_-(y)$ and equation 41 follows. $\square$

**Remark.** Theorem C.2 shows that, in the population limit, a weighted logistic oracle estimates a (weighted) log-odds function. This makes exponential tilting with $S(y; d^\star) = d^\star f(y)$ a principled way to bias generation toward one directional class.

## C.4 Weighted Denoising Cross-Entropy Fits a Target Distribution

We formalize the effect of WDCE as fitting an MDLM to a reweighted data distribution.

**Lemma C.1** (Weighted risk equals unweighted risk under a reweighted distribution). *Let $r$ be any distribution on $\mathcal{Y}$ and let $w : \mathcal{Y} \to [0, \infty)$ be a weight function with $0 < \mathbb{E}_{Y \sim r}[w(Y)] < \infty$. Define the normalized reweighted distribution*

$$\pi(y) := \frac{r(y)w(y)}{\mathbb{E}_{Y \sim r}[w(Y)]}. \tag{44}$$

*Then for any nonnegative loss $\ell(y)$,*

$$\mathbb{E}_{Y \sim r}[w(Y)\ell(Y)] = \mathbb{E}_{Y \sim r}[w(Y)] \cdot \mathbb{E}_{Y \sim \pi}[\ell(Y)]. \tag{45}$$

*Proof.* By definition, $\mathbb{E}_{Y \sim \pi}[\ell(Y)] = \sum_y \pi(y)\ell(y) = \frac{1}{\mathbb{E}_r[w]} \sum_y r(y)w(y)\ell(y)$. Rearrange. $\square$

**Theorem C.3** (Population optimality of WDCE denoisers). *Fix a corruption kernel family $q_t(x_t \mid x)$ and a time sampling distribution over $t \in [0, 1]$. Let $\pi$ be a target distribution on $\mathcal{Y}$ and define the joint $(X, X_t)$ by $X \sim \pi$ and $X_t \sim q_t(\cdot \mid X)$. Consider the MDLM denoising objective*

$$\mathcal{L}_\pi(\theta) = \mathbb{E}_{t,X,X_t}\left[ \sum_{\ell:(X_t)_\ell=[\text{MASK}]} -\log p_\theta(X_\ell \mid (X_t)_{\text{UM}}, t) \right]. \tag{46}$$

*Then for each time $t$ and each masked position $\ell$, the minimizer satisfies*

$$p_{\theta^\star}(X_\ell = a \mid (X_t)_{\text{UM}}, t) = \mathbb{P}_\pi(X_\ell = a \mid (X_t)_{\text{UM}}, t) \qquad \forall a \in \mathcal{A}, \tag{47}$$

*that is, the optimal denoiser recovers the true conditional under $\pi$.*

*Proof.* Fix $t$ and condition on the context $C := ((X_t)_{\text{UM}}, t)$ and the event that position $\ell$ is masked. The inner term in equation 46 is the cross-entropy between the true conditional distribution of $X_\ell$ given $C$ and the model distribution $p_\theta(\cdot \mid C)$. Cross-entropy is minimized uniquely by matching the true conditional. Taking expectation over contexts yields the result. $\square$

**Connection to reward tilting.** If the proposal distribution $r$ is $p_0$ and weights are $w(y) = \exp(S(y)/\alpha)$, then the reweighted distribution $\pi$ in Lemma C.1 equals the tilted distribution $p^\star$ in equation 32. Thus WDCE is (in the population limit) standard MDLM training under the tilted target distribution.

## C.5 Stability of Tilting Under Oracle Approximation Error

**Theorem C.4** (Tilt robustness under bounded score error). *Let $S^\star : \mathcal{Y} \to \mathbb{R}$ be an ideal score and let $S : \mathcal{Y} \to \mathbb{R}$ satisfy*

$$\sup_{y \in \mathcal{Y}} |S(y) - S^\star(y)| \le \varepsilon. \tag{48}$$

*Let $p^\star$ and $\widetilde{p}$ be the corresponding tilted distributions built from $(p_0, S^\star)$ and $(p_0, S)$ with the same temperature $\alpha > 0$. Then*

$$\text{KL}(\widetilde{p}\|p^\star) \le \frac{2\varepsilon}{\alpha}, \qquad \text{KL}(p^\star\|\widetilde{p}) \le \frac{2\varepsilon}{\alpha}, \tag{49}$$

*and therefore, by Pinsker's inequality,*

$$\|\widetilde{p} - p^\star\|_{\text{TV}} \le \sqrt{\frac{\varepsilon}{\alpha}}. \tag{50}$$

*Proof.* Write $S = S^\star + \delta$ with $|\delta(y)| \le \varepsilon$. Then

$$\widetilde{p}(y) = \frac{p_0(y)\exp((S^\star(y) + \delta(y))/\alpha)}{\widetilde{Z}} = p^\star(y)\,\frac{\exp(\delta(y)/\alpha)}{\mathbb{E}_{Y \sim p^\star}[\exp(\delta(Y)/\alpha)]}. \tag{51}$$

Since $\exp(\delta/\alpha) \in [\exp(-\varepsilon/\alpha), \exp(\varepsilon/\alpha)]$, the normalizer ratio satisfies

$$\mathbb{E}_{p^\star}[\exp(\delta/\alpha)] \in [\exp(-\varepsilon/\alpha), \exp(\varepsilon/\alpha)]. \tag{52}$$

Hence for all $y$,

$$\log\frac{\widetilde{p}(y)}{p^\star(y)} = \frac{\delta(y)}{\alpha} - \log\mathbb{E}_{p^\star}[\exp(\delta/\alpha)] \in \left[-\frac{2\varepsilon}{\alpha}, \frac{2\varepsilon}{\alpha}\right]. \tag{53}$$

Taking expectation under $\widetilde{p}$ yields $\mathrm{KL}(\widetilde{p}\|p^\star) \le 2\varepsilon/\alpha$. The reverse KL bound follows symmetrically by swapping roles of $(S, S^\star)$. Pinsker's inequality gives the TV bound. □

## C.6 A SEPARABILITY CONSEQUENCE OF ZERO CONTRASTIVE LOSS

**Proposition C.2** (Zero margin-contrastive loss implies linear separability). *Let $\{(y_i, d_i)\}_{i=1}^N$ be labeled samples with $d_i \in \{+1, -1\}$ and embeddings $h(y_i) \in \mathbb{R}^m$. Consider the margin-contrastive loss*

$$\mathcal{L}_{\mathrm{ctr}} = \sum_{(i,j):d_i=d_j} \|h(y_i) - h(y_j)\|_2^2 + \sum_{(i,j):d_i \ne d_j} \max(0,\, m_0 - \|h(y_i) - h(y_j)\|_2)^2. \tag{54}$$

*If $\mathcal{L}_{\mathrm{ctr}} = 0$, then there exist $u_+, u_- \in \mathbb{R}^m$ such that $h(y_i) = u_{d_i}$ for all $i$ and $\|u_+ - u_-\|_2 \ge m_0$. In particular, the classes are linearly separable by a hyperplane with margin at least $m_0/2$.*

*Proof.* If $\mathcal{L}_{\mathrm{ctr}} = 0$, then for any pair $(i, j)$ with $d_i = d_j$ we must have $\|h(y_i) - h(y_j)\|_2^2 = 0$, hence all embeddings within a class are identical. Denote the two class prototypes by $u_+$ and $u_-$. For any pair with $d_i \ne d_j$, the hinge term being zero implies $\|u_+ - u_-\|_2 \ge m_0$.

Define $w := u_+ - u_-$ and $b := -\frac{1}{2}\langle w, u_+ + u_-\rangle$. Then

$$\langle w, u_+\rangle + b = \tfrac{1}{2}\|u_+ - u_-\|_2^2 \ge \tfrac{1}{2}m_0^2, \qquad \langle w, u_-\rangle + b = -\tfrac{1}{2}\|u_+ - u_-\|_2^2 \le -\tfrac{1}{2}m_0^2, \tag{55}$$

so the hyperplane $\{z : \langle w, z\rangle + b = 0\}$ separates the two prototypes. The (geometric) margin is at least $\|u_+ - u_-\|_2/2 \ge m_0/2$. □

# D IMPLEMENTATION DETAILS

## D.1 DIRECTION ORACLE TRAINING

The Direction Oracle is trained for 20 epochs using AdamW optimization with a learning rate of $10^{-5}$ and batch size of 16, minimizing cross-entropy loss. pre-trained encoders remain frozen throughout training; only projection layers, self-attention and cross-attention modules, and the two-layer MLP classifier head are optimized. Due to the limited size of available labeled data, we train on the full training split without validation and evaluate performance exclusively on an independent held-out test set.

## D.2 TD3B FINETUNING

For tree search-based sampling, we employ trajectory-aware tree search with 20 iterations and 24 children per node, sampling 4 targets per iteration with a buffer size of 32 candidates per target. A replay buffer of 2000 samples with FIFO replacement is maintained to mitigate catastrophic forgetting. Training uses a batch size of 4 with gradient accumulation over 4 steps, a learning rate of $5 \times 10^{-5}$, and 4 WDCE replicates per sample. The KL regularization coefficient $\lambda_{\text{reg}}$ is set to 0.5, and tree search resampling is performed every 10 epochs. Training is conducted on 8 NVIDIA A100 GPUs using PyTorch DDP with synchronized buffer aggregation.

# E CASE STUDIES ON ADDITIONAL PROTEIN TARGETS

We further applied TD3B to TAAR1, a neuromodulatory GPCR implicated in dopaminergic and serotonergic signaling and strongly linked to schizophrenia, where both agonists and antagonists are of pharmacological interest (Dedic et al., 2021). As shown in Figure E1, TD3B-generated agonists and antagonists for TAAR1 exhibit distinct binding modes. Similar to the activatory contacts of T1AM (3-iodothyronamine), a small molecule TAAR1 agonist shown in PDB 8JLN (Figure E1A) (Xu et al., 2023), the designed agonist preferentially engages with a compact orthosteric pocket spanning TM3, TM5, TM6, and TM7, Jones et al. (2020) including residues associated with receptor activation (Huang et al., 2025) (Figure E1B), and the designed antagonist occupies a broader binding region that partially overlaps the orthosteric site but selectively avoids key activation-associated transmembrane contacts (Figure E1C).

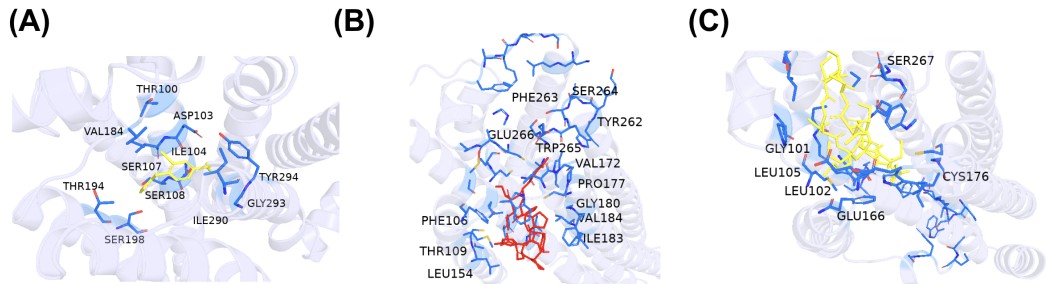

Figure E1: Evaluation of TD3B on TAAR1. **(A)** Existing TAAR1 agonist bound to TAAR1. **(B)** TD3B-designed agonist bound TAAR1. **(C)** TD3B-designed antagonist bound TAAR1.

We next applied TD3B to the Orexin 1 Receptor (OX1R), a regulatory GPCR where antagonists treat insomnia and agonists show promise for narcolepsy (Scammell & Winrow, 2011; Nishino et al., 2000). Figure E2A shows the crystal structure of OX1R bound to the clinical antagonist suvorexant (PDB: 4ZJ8), revealing 14 key orthosteric binding site residues. TD3B-designed agonists engage 5 conserved residues, while antagonists engage 6 residues (Figures E2B and C). Critically, both ligands interact with GLN126, a molecular switch that controls receptor activation through hydrogen bonding with TYR348 (Karhu et al., 2019). In antagonists, the GLN126-TYR348 bond stabilizes the inactive state, while agonists are predicted to disrupt this interaction to enable activation (Yin et al., 2016). The substantial conservation with suvorexant's validated binding site confirms orthosteric targeting

and demonstrates TD3B's ability to design functionally distinct ligands for the same receptor pocket. Together, these results indicate that TD3B can generate functionally divergent binders for the same target by modulating transition directionality rather than binding affinity alone.

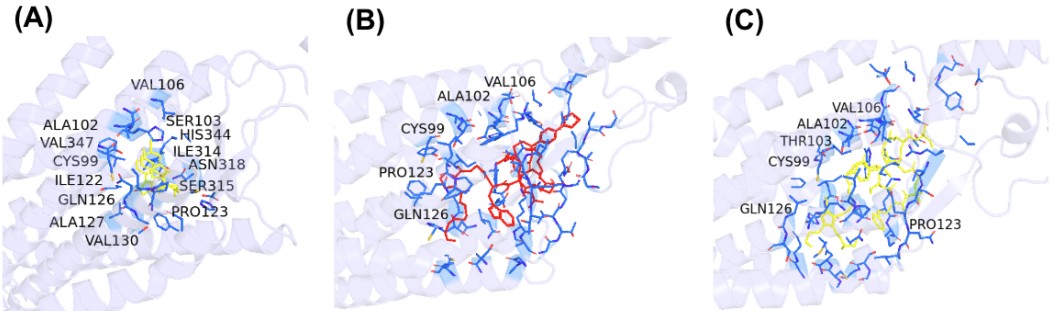

Figure E2: Evaluation of TD3B on OX1R. **(A)** Suvorexant bound to OX1 (PDB: 4ZJ8). **(B)** TD3B-designed agonist. **(C)** TD3B-designed antagonist.

## F    RELATED WORKS

### F.1    ALLOSTERIC BINDER DESIGN

Classical allosteric theory originated from the MWC concerted transition Monod et al. (1965) and KNF sequential induced-fit Koshland Jr et al. (1966) models, while early discovery relied on serendipitous screening hits Lu et al. (2019). With the expansion of structural databases Shen et al. (2016), computational methods emerged to predict allosteric sites Huang et al. (2013); Akbar & Helms (2018), hotspots Clarke et al. (2016), and communication pathways Tan et al. (2019); Halabi et al. (2009) from static structures. However, static approaches miss cryptic sites that remain occluded in certain conformations. Dynamics-based methods using MD simulations and Markov state models Shukla et al. (2014); Bowman et al. (2015) address this limitation but incur high computational costs.

*De novo* allosteric design has evolved from early Rosetta-based side-chain networks Churchfield et al. (2016); Pirro et al. (2020) to recent modular rigid-body coupling strategies leveraging RFDiffusion Watson et al. (2023) and ProteinMPNN Dauparas et al. (2022), enabling peptide-responsive ring architectures and effector-induced cage disassembly Pillai et al. (2024). Complementary approaches include structure-based design for GPCRs Li et al. (2025) and Chemical Language Model (CLM)-based generation Ballarotto et al. (2023). In contrast, TD3B reframes directional design as a non-equilibrium transport problem, introducing directional supervision such that binding agents act as one-way valves controlling state transition directionality.

### F.2    CONDITIONAL GENERATION FOR DISCRETE DIFFUSION

Diffusion models have emerged as powerful unsupervised generative frameworks capable of capturing distributions over discrete spaces, with conditional generation enabling efficient sample exploration Austin et al. (2021); Lou et al. (2024); Sahoo et al. (2024); Shi et al. (2024). To steer generation toward desired properties, guided generation approaches incorporate classifier gradients into the sampling process. These include training-free methods such as Classifier Guidance (CG) and Sequential Monte Carlo (SMC) Dhariwal & Nichol (2021); Nisonoff et al. (2024); Chung et al. (2022); Wu et al. (2023); Dou & Song (2024); Phillips et al. (2024), as well as Classifier-Free Guidance (CFG) Ho & Salimans (2022). While computationally efficient, these approaches struggle to provide complex guidance in discrete domains, where gradient-based steering is inherently limited by non-differentiable operations.

Reinforcement learning-based methods address this limitation by enabling discrete diffusion models to learn more sophisticated conditional distributions through environment interaction. Policy gradient approaches such as DRAKES and GLID$^2$E Wang et al. (2025); Cao et al. (2025) update model policies via reward signals, while tree-search methods including PepTune and TR2-D2 Tang et al. (2025a;c) offer greater flexibility for multi-condition and multi-objective sampling by explicitly exploring the combinatorial space.

